# Urinary prostaglandin E2 as a biomarker for recurrent UTI in postmenopausal women

Tahmineh Ebrahimzadeh[1], Amy Kuprasertkul[2], Michael L Neugent[1], Kevin C Lutz[3], Jorge L Fuentes[2], Jashkaran Gadhvi[1], Fatima Khan[1], Cong Zhang[3], Belle M Sharon[1], Kim Orth[4,5,6], Qiwei Li[3], Philippe E Zimmern[2], Nicole J De Nisco[1]

Urinary tract infection (UTI) is one of the most common adult bacterial infections and exhibits high recurrence rates, especially in postmenopausal women. Studies in mouse models suggest that cyclooxygenase-2 (COX-2)–mediated inflammation sensitizes the bladder to recurrent UTI (rUTI). However, COX-2–mediated inflammation has not been robustly studied in human rUTI. We used human cohorts to assess urothelial COX-2 production and evaluate its product, $PGE_2$, as a biomarker for rUTI in postmenopausal women. We found that the percentage of COX-2–positive cells was elevated in inflamed versus uninflamed bladder regions. We analyzed the performance of urinary $PGE_2$ as a biomarker for rUTI in a controlled cohort of 92 postmenopausal women and $PGE_2$ consistently outperformed all other tested clinical variables as a predictor of rUTI status. Furthermore, time-to-relapse analysis indicated that the risk of rUTI relapse was 3.6 times higher in women with above median urinary $PGE_2$ levels than with below median levels. Taken together, these data suggest that urinary $PGE_2$ may be a clinically useful diagnostic and prognostic biomarker for rUTI in postmenopausal women.

## Introduction

Urinary tract infection (UTI) affects more than 150 million people annually worldwide (Stamm & Norrby, 2001; Flores-Mireles et al, 2015). UTIs are a leading indication for prescription antibiotics and have a significant impact on women of all ages (Amna et al, 2013). Recurrent urinary tract infection (rUTI) is defined as ≥3 symptomatic UTIs in 12 mo or ≥2 symptomatic UTIs in 6 mo (Nicolle, 2001; Glover et al, 2014; Malik et al, 2018b). rUTI has a severe impact on quality of life causing pain, frequency, urgency, anxiety, and symptoms of clinical depression among affected women (Ellis & Verma, 2000; Renard et al, 2014). rUTI incidence increases with age with reported recurrence rates as high as 50% in postmenopausal women (Foxman, 2002, 2014; Raz, 2011). rUTI is caused by diverse bacterial species and less commonly by

fungi (Flores-Mireles et al, 2015). The front-line therapy for rUTI is prescription of antibiotics such as trimethoprim sulfamethoxazole, nitrofurantoin, and fluoroquinolones (Jancel & Dudas, 2002). Resistance, allergy, or documented adverse reactions to antibiotics are frequent in the elderly and limit the safety and efficacy of antibiotics in this demographic (Jancel & Dudas, 2002; Malik et al, 2018a). Therefore, new therapies for rUTI must be developed.

rUTI is, in part, an inflammatory disease (Bjorling et al, 2011). Evidence of chronic inflammation has been reported in the bladders of rUTI patients during cystoscopy (Wu et al, 2016; Crivelli et al, 2019). Extensive work performed exclusively in mouse models has implicated host inflammation, specifically Cyclooxygenase-2 (COX-2)–mediated inflammation, as a key sensitizing factor for rUTI (Hannan et al, 2010; Hannan et al, 2014; O'Brien et al, 2016). Excessive urothelial neutrophil infiltration and COX-2–dependent inflammation were found to cause tissue damage and remodeling that sensitize the bladder to severe rUTI (Hannan et al, 2014; O'Brien et al, 2015; O'Brien et al, 2016). Importantly, Hannan et al (2014) demonstrated that treatment of mice with specific COX-2, but not COX-1, inhibitors protected the bladder against sensitization to severe rUTI (Hannan et al, 2014). Furthermore, treatment of mice with dexamethasone, another drug targeting the COX-2 pathway, suppressed development of chronic cystitis (Hannan et al, 2010). These data suggest that the COX-2 pathway may be a promising therapeutic target for rUTI; however, knowledge of the role of COX-2–mediated inflammation in human rUTI is limited.

Here, we used defined, well-curated human cohorts to evaluate activation of the COX-2–mediated inflammation during rUTI in postmenopausal women. We hypothesized that, as observed in mouse models of UTI, COX-2 would be expressed in urothelium of visibly inflamed bladder regions in human rUTI patients. To determine if COX-2 enzyme levels are elevated in regions of cystitis, we enumerated COX-2–expressing urothelial cells in bladder biopsies from inflamed and control regions. We assessed urinary $PGE_2$ as a proxy for urothelial COX-2 expression in matched bladder biopsy and urine samples. We then evaluated $PGE_2$ as a biomarker for rUTI in postmenopausal women by measuring urinary $PGE_2$ levels across three groups with different UTI histories. Finally, we

[1]Department of Biological Sciences, University of Texas at Dallas, Richardson, TX, USA  [2]Department of Urology, University of Texas Southwestern Medical Center, Dallas, TX, USA  [3]Depatment of Mathematics, University of Texas at Dallas, Richardson, TX, USA  [4]Department of Molecular Biology, University of Texas Southwestern Medical Center, Dallas, TX, USA  [5]Howard Hughes Medical Institute, University of Texas Southwestern Medical Center, Dallas, TX, USA  [6]Department of Biochemistry, University of Texas Southwestern Medical Center, Dallas, TX, USA

Correspondence: nicole.denisco@utdallas.edu

performed a time-to-relapse study to determine if urinary PGE$_2$ levels were predictive of rUTI relapse.

# Results

### COX-2 expression is activated in the bladder urothelium during human rUTI

Previous work in mouse models has demonstrated that COX-2 expression triggers prolonged neutrophil recruitment resulting in tissue damage, changes to bladder wall morphology, and increased susceptibility to severe rUTI (Hannan et al, 2014; O'Brien et al, 2016; Yu et al, 2019). To determine if COX-2 expression was activated during human rUTI, we analyzed COX-2 expression and neutrophil recruitment in bladder urothelium in two cohorts of women undergoing cystoscopy with fulguration of trigonitis (CFT). For Cohort 1, antibiotic therapy was stopped after symptom resolution 7 d before CFT, and two biopsies, one from a visibly inflamed (I1) and one from a visibly normal control region (C1) of the bladder, were obtained (Fig 1A) (De Nisco et al, 2019). To determine if COX-2 expression was associated with neutrophil recruitment, we visualized COX-2 and

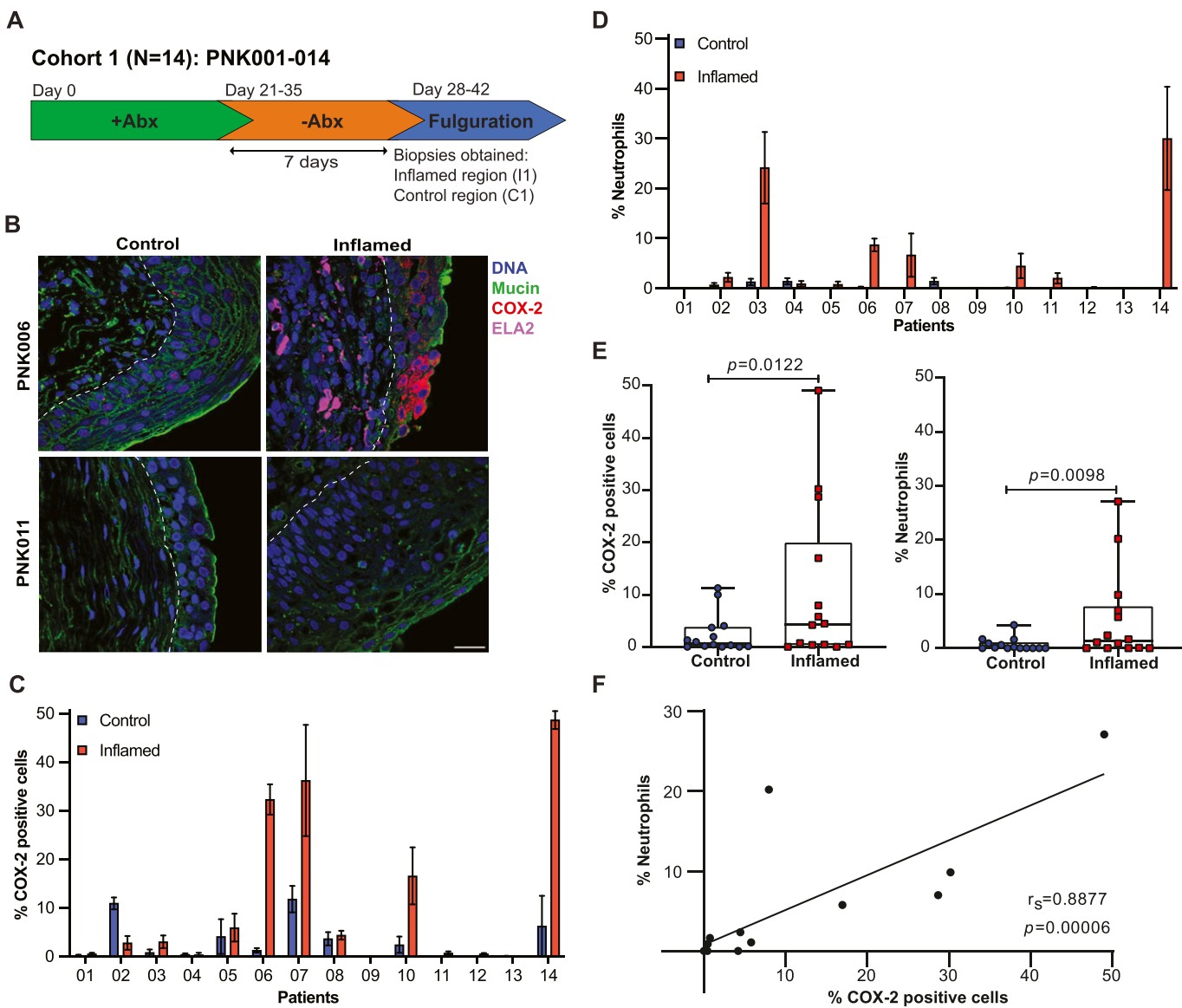

**Figure 1.   COX-2 expression is activated in the bladder urothelium during human recurrent urinary tract infection.**
**(A)** Cohort 1 patient recruitment and procedure timeline. **(B)** Representative confocal micrographs (63×) of I1 and C1 regions of PNK006 and PNK011 with DNA (Hoechst) in blue, Mucin (WGA) in green, COX-2 in red and neutrophils (ELA-2) in magenta. Scale bar represents 10 $\mu$m. **(C, D)** Quantification of COX-2 expressing cells and (D) neutrophils within the urothelium of control (blue) and inflamed (red) region biopsies reported as percentage of total urothelial cells. 10 randomly sampled images were enumerated for each section. Bar graphs represent mean ± SEM. **(E)** Comparison of %COX-2–expressing cells and %neutrophils between control and inflamed regions. Whiskers drawn minimum to maximum, boxes represent interquartile range, and median denoted by horizontal line. *P*-values generated by Wilcoxon matched pairs signed-ranks test. **(F)** Linear regression with Spearman correlation between %neutrophils and %COX-2–positive cells.

elastase (ELA2), a neutrophil marker (Lammers et al, 1986), in the I1 and C1 regions using immunofluorescence (IF) confocal microscopy (Figs 1B and S1). COX-2–positive urothelial cells were not observed in every I1 biopsy. Representative images of I1 biopsies with high urothelial COX-2 (PNK006) versus undetectable urothelial COX-2 (PNK011) are presented in Fig 1B. Quantification of COX-2–expressing urothelial cells and neutrophil infiltration was performed and reported as percentage of total of urothelial cells (Fig 1C and D). Urothelial COX-2 expression was observed in the I1 regions of 85.7% (12/14) of patients (Fig 1B and C). Similarly, neutrophil infiltration was observed in the I1 regions of 71.4% (10/14) of patients (Fig 1B and D). The quantification data for COX-2 and neutrophils were summarized and classified into two groups: inflamed and control. The median percentage of urothelial COX-2–expressing cells and neutrophils was 6.1 and 24.1 times higher in inflamed versus control regions, respectively (Fig 1E). Correlation analysis between neutrophil infiltration and COX-2–expressing cells in the I1 area revealed a strong correlation between the two inflammatory markers ($r_S$ = 0.8877, $P$ = 0.00006) (Fig 1F). In these biopsies we defined the suburothelium as the region underlying the urothelium that includes the lamina propria and in some cases the muscularis propria. Suburothelial neutrophil accumulation was found in patients PNK003, PNK006, PNK010, PNK011, and PNK014, indicating that neutrophil accumulation was not limited to the urothelium (Fig S2A–E). Suburothelial COX-2 expression was also observed but predominately in regions with severely damaged urothelium or co-localized with neutrophils (Fig S2A, C, E, and F).

## Urinary PGE$_2$ as a marker for COX-2–mediated bladder inflammation

We next sought to identify a marker of COX-2 activity that would be measurable in urine. Prostaglandin E2 (PGE$_2$) is the product of arachidonic acid conversion by the COX-2 enzyme (Ricciotti & FitzGerald, 2011). Extracellular PGE$_2$ elicits diverse cellular responses including cell proliferation, angiogenesis, pain sensation, and inflammation (Nakanishi & Rosenberg, 2013; Kawahara et al, 2015). Induced COX-2 expression results in higher secreted levels of PGE$_2$ (Wheeler et al, 2002; Park et al, 2006). Therefore, we hypothesized that urinary PGE$_2$ should be a reliable indicator of urothelial COX-2 expression (Wheeler et al, 2002). To test this hypothesis, we measured both urothelial COX-2 expression and urinary PGE$_2$ in a second CFT cohort (Cohort 2, PNK016-27) (Fig 2A). For this cohort, due to changes in the IRB-approved protocol, antibiotic therapy was not ceased before CFT and one inflamed-region biopsy (I1) was obtained. Cohort 2 patient statistics and clinical urine culture (UC) results are reported in Table 1.

We performed IF for COX-2 and neutrophils on I1 bladder biopsy tissues and commercially available normal bladder biopsy sections as a control (US Biomax). Representative images of control, PNK025 and PNK027 biopsy sections are shown in Figs 2B and S3. COX-2 expressing urothelial cells or neutrophils were counted and reported as percentage to total number of urothelial cells (Fig 2C and D). PNK025 and PNK027 biopsy sections showed the highest median percentages of COX-2 expressing cells (16.5% and 33.5%, respectively) (Fig 2C). Urothelial neutrophil infiltration was found in PNK016 and PNK023-27 but was highest in PNK025 (median = 21.9%)

and PNK027 (median = 18.5%), mirroring the high percentage of COX-2–expressing cells detected in these biopsies (Fig 2D).

We next measured urinary PGE$_2$ levels. To control for differences in urine concentration, urinary biomarkers are often normalized to creatinine (Cr). Urinary creatinine excretion rate is assumed to remain constant within and between individuals; however, recent reports indicate that urinary creatinine excretion rate may vary widely between individuals depending on different clinical factors (Tang et al, 2015). Accordingly, Tang et al (2015) suggest both normalized and non-normalized data should be reported. Urinary PGE$_2$ concentration is presented normalized to creatinine (Fig 2E) and in raw values (Fig 2F). To determine if urinary PGE$_2$ is associated with urothelial COX-2 expression, we performed linear regression analysis between PGE$_2$ concentration and percentage of COX-2-positive urothelial cells and observed a robust positive association between urinary PGE$_2$ concentration and percentage of urothelial COX-2–expressing cells ($r_S$ = 0.8818 $P$ = 0.0007) (Fig 2G). Urinary PGE$_2$ levels were highest in PNK016, 23, 25, and 27 (Fig 2F and G). Although the urinary PGE$_2$ concentration of PNK016 was high (2,670.7 pg/ml), the percentage of COX-2–positive urothelial cells and neutrophils was relatively low in the biopsied tissue (Fig 2C and D). Because the selection of inflamed region biopsied was based on visible cues, one possible explanation is that the inflamed region biopsied for this patient was not representative of the bladder as a whole and regions with high numbers of COX-2–positive cells or neutrophils were not captured in the biopsy. For this reason, analysis of PGE$_2$ concentrations in the urine, which is less prone to sampling bias, is important. Interestingly, three of the patients with high urinary PGE$_2$ concentrations (PNK016, 23, and 27) were the only patients in this cohort who presented with positive UC on the day of CFT (Table 1).

Although sample size is low, one possible interpretation is that urinary PGE$_2$ levels are associated with bacteriuria. We observed a single outlier, PNK025, which was UC negative. We hypothesized that although antibiotics had eliminated urinary bacteria, bacteria may still be present within the bladder wall. To test this hypothesis, we performed FISH with a 16S rRNA universal bacterial probe on all Cohort 2 biopsies (Neugent et al, 2019). Representative images demonstrate the presence of tissue-associated bacteria in the bladder wall of PNK025 and PNK027 (Fig 3A). Quantification of bacterial community size revealed the largest load of tissue-resident bacteria in PNK025 (Fig 3B), suggesting that although the UC for PNK025 was negative, there was a high burden of tissue-resident bacteria (Fig 3A and B). In accordance with previously published observations, we detected suburothelial bacterial communities in the bladder wall of several patients including PNK016-19, 22, 25, and 26 (Figs 3B and S4). Taken together, higher urinary PGE$_2$ levels were associated with urothelial COX-2 expression as well as with a higher bacterial load either in the urine or bladder wall.

## PGE$_2$ is elevated during rUTI relapse but not during remission

Previous studies have shown that expression of the gene encoding COX-2, *Ptgs2*, was elevated 24 h postinfection in rUTI-sensitized mice but did not remain elevated during convalescence (Hannan et

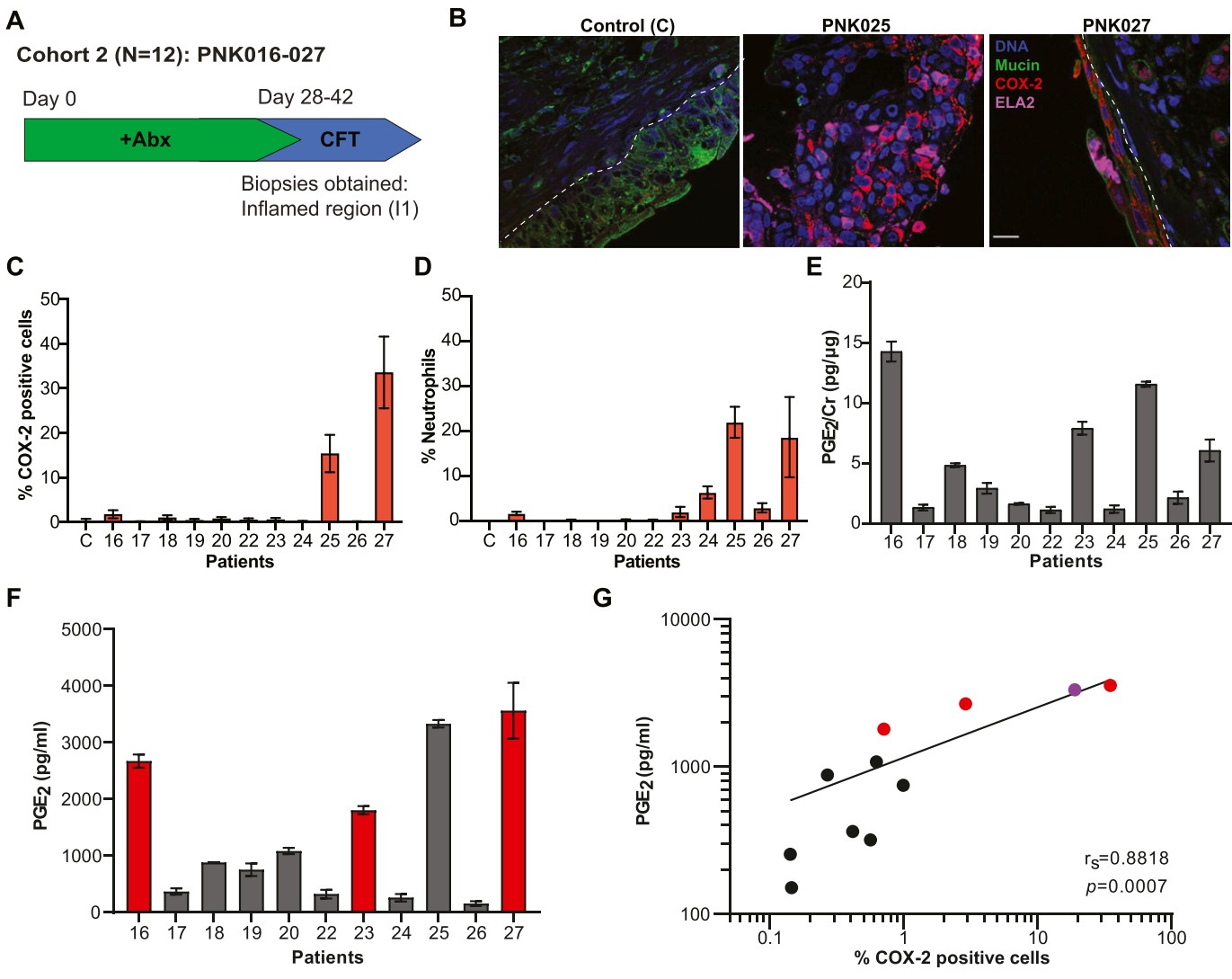

**Figure 2.  Urinary PGE₂ as a marker for COX-2–mediated bladder inflammation.**
**(A)** Cohort 2 patient recruitment and procedure timeline. **(B)** Representative confocal micrographs for control and I1 region from patients PNK025 and PNK027 compared to commercially available human normal bladder section (control, US Biomax) with DNA in blue, Mucin in green, COX-2 in red and neutrophils in magenta. Scale bar represents 10 $\mu$m. **(C, D)** Quantification of COX-2 expressing cells and (D) neutrophils within the urothelium reported as percentage of total urothelial cells. 10 randomly sampled images taken with a 63× objective were enumerated for scoring. C denotes control. **(E)** Bar graphs represent mean ± SEM (E) Urinary PGE₂ normalized to Cr. **(F)** Raw urinary PGE₂ concentration. Bar graphs represent mean ± SD. **(G)** Linear regression with Spearman's correlation between raw urinary PGE₂ concentration and %COX-2–positive urothelial cells. Red circle denotes positive urine culture. Purple circle denotes PNK025.

al, 2014). These data along with our observed association between urinary PGE₂ and bacterial load led us to hypothesize that urinary PGE₂ may be a biomarker for active rUTI in postmenopausal women. To test this hypothesis, we measured urinary PGE₂ in a third cohort (Cohort 3, n = 92) of postmenopausal women. rUTI patients typically oscillate between rUTI remission and relapse (Dason et al, 2011; Corriere et al, 2013). Accordingly, women were stratified into three groups based on their current infection status and rUTI history: Never (no UTI history), Remission (rUTI history, no current rUTI) and Relapse (rUTI history, current UTI) (Fig 4A). The Never group served as a control for normal variations in urinary PGE₂ levels between individuals. The Remission group allowed determination of COX-2 activity in the absence of active infection in rUTI-sensitized

individuals. None of the women in Cohort 3 were undergoing CFT and active rUTI in the Relapse group was managed by antibiotic therapy. Cohort summary statistics are reported in Table 2.

We observed significantly elevated urinary PGE₂ concentration, both raw and normalized to Cr, in Relapse patients as compared to the Remission and Never patients (Fig 4B and C). The median level of PGE₂ in the Relapse group was 2,318 pg/ml (Interquartile range [IQR]: 1,859–2,879) compared to 1,050 pg/ml (IQR: 464.5–1,675) and 965.1 pg/ml (IQR: 391.9–1,773) in the Remission and Never groups, respectively (Fig 4C). We observed no significant difference in urinary PGE₂ levels between the Remission and Never groups (Fig 4C). These data suggest that during rUTI remission urinary PGE₂

**Table 1. Cohort 2 patient characteristics.**

| Patients | Age (yr) | BMI (kg/m²) | Diabetes | Prior CFT | Urine culture history | Urine culture before CFT |
|---|---|---|---|---|---|---|
| PNK016 | 58 | 32.2 | IDDM | No | *Pseudomonas aeruginosa* and *Proteus mirabilis* | *Enterococcus faecalis* ($10^5$) |
| PNK017 | 65 | 25.3 | No | No | *P. aeruginosa*, *Escherichia coli*, and *E. faecalis* | No growth |
| PNK018 | 62 | 24.0 | No | No | *Klebsiella pneumoniae*, *E. coli*, and *E. faecalis* | No growth |
| PNK019 | 80 | 22.5 | No | No | *E. faecalis* and *K. pneumoniae* | No growth |
| PNK020 | 51 | 31.9 | No | No | *E. coli* | No growth |
| PNK022 | 88 | 27.5 | No | No | *E. coli* | No growth |
| PNK023 | 82 | 20.7 | No | No | *K. pneumoniae* | *K. pneumoniae* ($10^5$) |
| PNK024 | 58 | 30.9 | AODM | Yes | *K. pneumoniae* | No growth |
| PNK025 | 66 | 41.2 | No | No | *E. coli* | No growth |
| PNK026 | 65 | 35.0 | AODM | No | *E. coli* | No growth |
| PNK027 | 77 | 22.0 | No | Yes | *E. coli* and *E. faecalis* | *Enterococcus faecium* ($10^5$), *Aerococcus urinae* (55–99,000) |

Relevant patient data recorded for the 11 study participants. BMI, body mass index. Diabetes: no, nondiabetic; IDDM, diabetes mellitus type 1; AODM, adult-onset diabetes mellitus type 2; CFT, cystoscopy with fulguration of trigonitis. Urine cultures performed in clinical laboratories with a $10^4$ CFU/ml detection limit.

returns to basal levels; however, longitudinal studies must be conducted to confirm this result.

### PGE₂ is a biomarker of active rUTI in postmenopausal women

We then analyzed cohort-associated clinical metadata to identify additional variables associated with group (Never, Remission, Relapse) membership or rUTI status. A total of 16 variables were tested including age, BMI, urine pH, estrogen hormone therapy (EHT), prolapse stage, post void residual (PVR) and diabetes (Tables S1 and S2). Besides urinary $PGE_2$ concentration ($P < 0.001$) and $PGE_2$/

Cr ratio ($P < 0.001$), BMI was the only variable significantly associated with group membership ($P = 0.0161$, Fig S5A). $PGE_2$ concentration, $PGE_2$/Cr ratio, and BMI were positively associated with active rUTI, whereas EHT ($P = 0.01246$) was negatively associated with active rUTI (Table S1 and Fig S5B). We next used logistic regression to detect biomarkers of rUTI status within the clinical metadata (Fig S6). The $PGE_2$ logistic regression model outperformed all other single-variable models in predicting rUTI status with an area under the curve (AUC) of 0.841 (Fig 4D and Table 3). Adding the covariates of age, BMI, EHT, PVR, and urine pH improved the model slightly (AUC = 0.880) but $PGE_2$ was the primary driver of the model

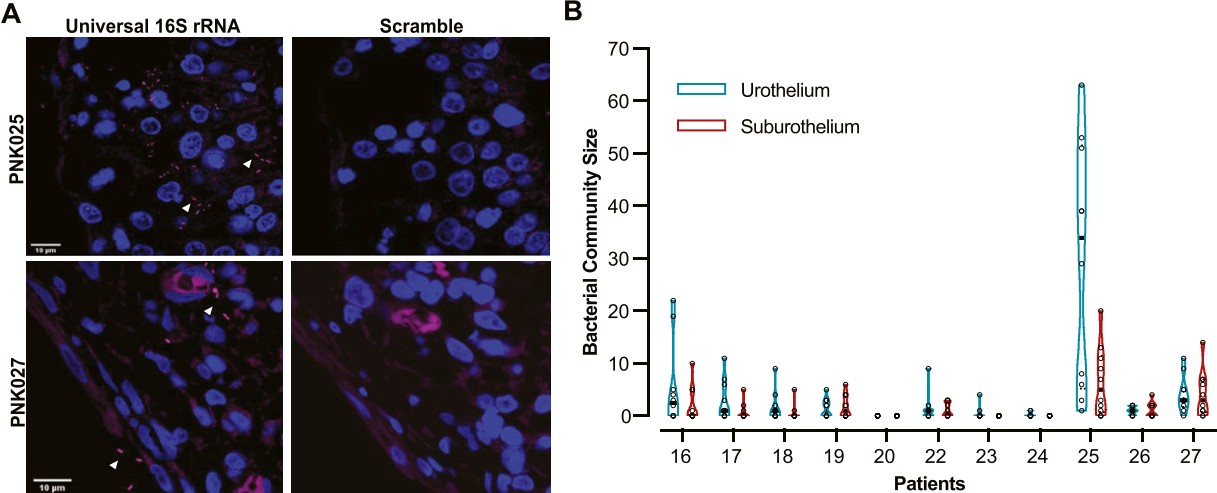

**Figure 3. FISH detects bladder-resident bacterial communities.**
**(A)** Representative confocal micrograph of FISH performed on PNK025 and PNK027 I1 biopsies using universal 16S rRNA and scramble probes with bacteria in magenta and DNA in blue. White arrowheads point to bacteria. **(B)** Violin plot of urothelial and suburothelial bacterial community enumeration in each biopsy. 10 randomly sampled images were quantified per biopsy. Individual data points are open circles and black boxes depict the median.

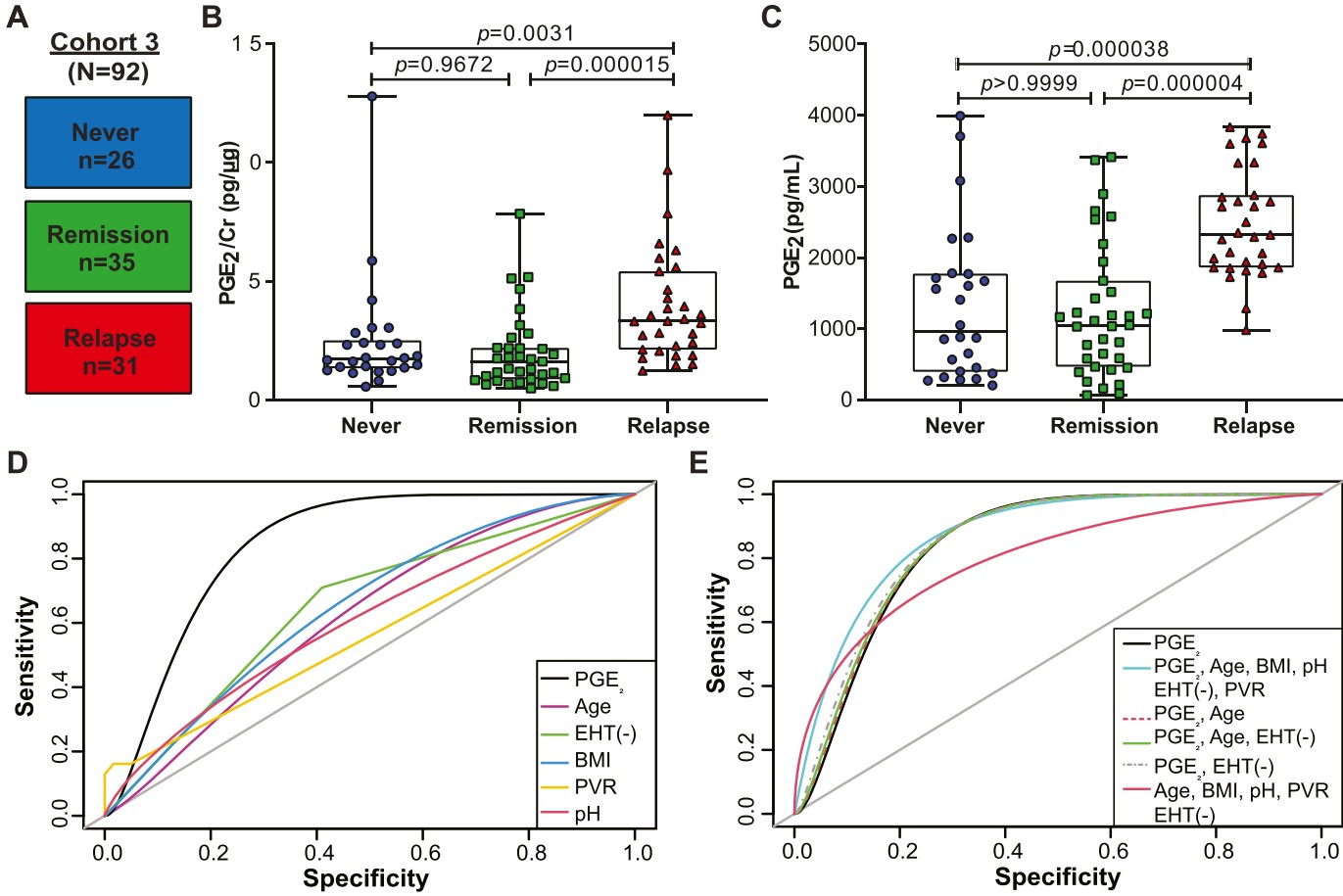

**Figure 4. Urinary PGE₂ is a biomarker for active recurrent urinary tract infection (rUTI) and is the best model to predict rUTI relapse in postmenopausal women.**
**(A)** Diagram depicting Cohort 3 design. Patients were stratified intro three cohorts: Never (no clinical history of urinary tract infection [UTI]), Remission (history of rUTI, no current UTI), and Relapse (history of rUTI, current UTI). **(B, C)** Urinary PGE₂ normalized to creatinine and (C) raw urinary PGE₂ concentration as measured by ELISA. *P*-values generated by Kruskal-Wallis test and Dunn's multiple comparison test. Whiskers drawn to minimum and maximum, boxes represent interquartile range, and median is denoted by a horizontal line. **(D)** Receiver operating characteristic curves of single-variable logistic regression models illustrate the ability of each model to predict rUTI status. Area under the curve was calculated using leave-one-out cross validation. **(E)** Receiver operating characteristic curves comparing multivariable logistic regression models to a PGE₂-only model for predicting rUTI. Area under the curve was calculated using leave-one-out cross validation.

as dropping PGE₂ from the model reduced the AUC to 0.805 (Fig 4E and Table 3). Taken together, these results indicate that urinary PGE₂ is a strong predictor, or biomarker, of rUTI status in this cohort of postmenopausal women.

### PGE₂ is associated with active rUTI in postmenopausal women with adult-onset diabetes mellitus (AODM)

There is a high frequency (22–33%) of AODM in adults over the age of 65 years in the United States and a high incidence of rUTI in individuals with AODM (Kirkman et al, 2012; Corriere et al, 2013; Nitzan et al, 2015). Therefore, a clinically useful biomarker for rUTI would differentiate patients independent of AODM status. We identified Cohort 3 individuals with AODM and divided patients into groups defined by documented AODM diagnosis and UTI status at the time of urine donation: No active UTI and nondiabetic, Active UTI and nondiabetic, No active UTI and diabetic, and Active UTI and diabetic. Urinary PGE₂ levels were significantly elevated in the Active UTI versus No active UTI in both diabetic and nondiabetic groups (Fig

5A). These results suggest that urinary PGE₂ can be used as a biomarker of rUTI in individuals with and without AODM.

### Association between urinary PGE₂ and relevant clinical variables

COX-2–mediated inflammation can be affected by many clinically relevant factors, such as, age, BMI, and the common use of non-steroidal anti-inflammatory drugs (NSAIDs). In contrast to non-selective NSAIDs like ibuprofen, COX-2–selective inhibitors (e.g., Celecoxib) preferentially block COX-2 activity over COX-1 activity and are prescribed to adults for the management of osteoarthritis, rheumatoid arthritis, and acute pain (Schattenkirchner, 1997; Zarghi & Arfaei, 2011; Zweers et al, 2011). We found that 25% of women in Cohort 3 used nonselective or selective NSAIDs (Table 2). To determine if there was any association between NSAID use on urinary PGE₂, we classified Cohort 2 and 3 patients into three groups based upon NSAID use: no NSAID, nonselective NSAID and selective NSAID. We observed no statistically significant difference in urinary PGE₂ concentrations between these groups (Fig S7A). The distribution of

**Table 2.   Cohort 3 summary statistics.**

| | | Never | Remission | Relapse |
|---|---|---|---|---|
| Number of women (n) | | 26 | 35 | 31 |
| Hx of fulguration | | 0 | 20 | 19 |
| Median Age (IQR) (yr) | | 69.5 (63–77.75) | 70 (63–81) | 74 (70–80) |
| Median BMI (IQR) (kg/m$^2$) | | 26.05 (21.4–27.5) | 27.8 (22–31) | 27.9 (25.4–32.1) |
| Median urine pH (IQR) | | 6 (5–7) | 5.82 (5–6.74) | 5.35 (5–6) |
| AODM (%) | | 1 (3%) | 7 (20%) | 6 (19.35%) |
| EHT (%) | | 14 (53%) | 22 (62%) | 9 (29%) |
| NSAID | Selective (%) | 2 (7.69%) | 5 (14.28%) | 1 (3.22%) |
| | Nonselective (%) | 6 (23.07%) | 7 (20%) | 2 (6.45%) |

Median and interquartile range (IQR) for age; (BMI) body mass index, and urinary pH. AODM, adult-onset diabetes mellitus; EHT, estrogen hormone therapy; NSAID, nonsteroidal anti-inflammatory drugs.

specific NSAIDs used by patients in Cohorts 2 and 3 is reported in Fig S7B.

EHT has been shown to affect local immune responses (Porter et al, 2001). Since EHT is common among postmenopausal women (Stanton et al, 2020), we evaluated the association between urinary PGE$_2$ and EHT. Although no association was found between EHT and urinary PGE$_2$ concentration, only 29% of patients in the Relapse group used EHT compared to 62% and 53% of patients in the Remission and Never groups, respectively, (Fig S7C and Table 3). Similarly, no association was found between urinary PGE$_2$ and urine pH (Fig S7D), BMI (Fig S7E), and age (Fig S7F).

### Urinary PGE$_2$ concentration is predictive of rUTI relapse

To further investigate the association between elevated urinary PGE$_2$ and rUTI, we recorded time to relapse over a 12-mo period for the Relapse group. We dichotomized patients about the median PGE$_2$ concentration into above median (n = 15, >2,318 pg/ml) and below median (n = 16, ≤2,318 pg/ml) groups. Proportional hazard analysis using the Mantel–Cox log-rank test indicated a much

higher likelihood of rUTI relapse (HR$_{high/low}$ = 3.61, HR$_{low/high}$ = 0.277 $P$ = 0.0087) in the above median group compared to the below median group (Fig 5B). At the end of the follow-up period, 12/15 patients in the above median group experienced rUTI relapse versus 5/16 patients in the below median group (Fig 5B). These data suggest that urinary PGE$_2$ concentration is predictive of rUTI relapse in this cohort of postmenopausal women.

## Discussion

As the human population ages and antimicrobial resistance becomes more widespread, rUTI is becoming both more prevalent and more difficult to manage. Because development of new antibiotics has not kept pace with the rate of emergence of resistant organisms, alternate therapies that improve upon existing antibiotics must be developed (Gupta et al, 1999; Roca et al, 2015). One strategy is to encourage a more productive immune response by controlling excessive inflammation (Ingersoll & Albert, 2013). For rUTI, murine

**Table 3.   Summary of cohort-associated clinical metadata analysis to assess the model prediction accuracy.**

| Model variables | AUC | F score | Cutoff probability |
|---|---|---|---|
| PGE$_2$ | 0.843 | 0.778 | 0.317 |
| Age | 0.623 | 0.574 | 0.289 |
| Estrogen | 0.65 | 0.564 | 0.468 |
| BMI | 0.627 | 0.594 | 0.266 |
| Post void residual (PVR) | 0.56 | 0.504 | 0.299 |
| pH | 0.606 | 0.538 | 0.304 |
| PGE$_2$, age, BMI, estrogen, PVR, and pH | 0.882 | 0.761 | 0.308 |
| PGE$_2$ and age | 0.844 | 0.75 | 0.224 |
| PGE$_2$, age, and estrogen | 0.848 | 0.737 | 0.268 |
| PGE$_2$, and estrogen | 0.855 | 0.776 | 0.43 |
| Age, BMI, estrogen, PVR, and pH | 0.805 | 0.675 | 0.287 |

Leave-one-out cross-validation (LOOCV) procedure used to calculate the area under the ROC curve (AUC), F-score (predictive accuracy of the model), and the cutoff probability.

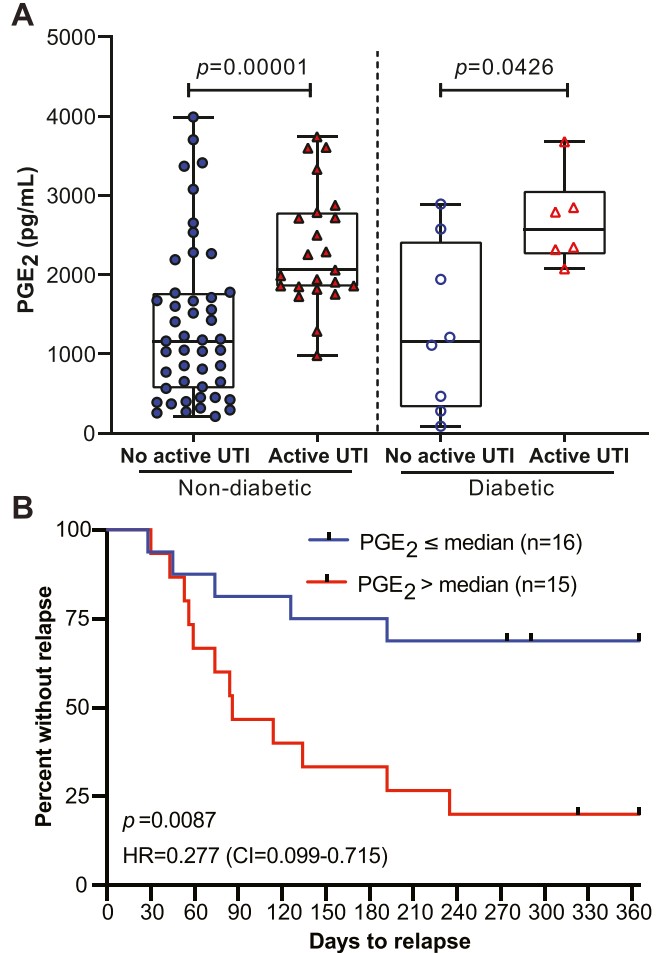

**Figure 5.** High urinary PGE$_2$ is a prognostic marker for development of recurrent urinary tract infection relapse in a 12-mo follow-up study. **(A)** Analysis of the urinary PGE$_2$ concentrations between nondiabetic (filled) and diabetic (empty) No active urinary tract infection (blue circle) versus Active urinary tract infection (red triangle) patients. Mann–Whitney U test used to calculate *P*-values. **(B)** Kaplan–Meier analysis of time-to-relapse data for patients in the Relapse group dichotomized about the median PGE$_2$ concentration. Red line depicts above median and blue line below median patients. Data were analyzed by log-rank (Mantel–Cox) test. HR, hazard ratio (below median/above median) and CI = 95% confidence interval.

studies have identified COX-2–mediated inflammation as a key sensitizing factor and a possible target for alternate rUTI therapies.

In this study, we demonstrate that the product of COX-2, PGE$_2$, is significantly elevated in the urine of postmenopausal women with active rUTI. Urinary PGE$_2$ concentration outperformed all other clinical variables as a predictor of rUTI status and was not significantly associated with clinical variables other than active rUTI. These findings suggest that urinary PGE$_2$ is a reliable biomarker for active rUTI in postmenopausal women. These observations are supported by previous reports of elevated urinary PGE$_2$ in young patients with UTI before antibiotic therapy (Wheeler et al, 2002). Importantly, we found that elevated PGE$_2$ concentration was strongly predictive of rUTI relapse in the studied cohort of postmenopausal women.

Previous clinical trials have evaluated the noninferiority of ibuprofen compared to antibiotic therapy, and, although over half

of the ibuprofen treatment cohort recovered without antibiotic therapy, the incidence of complications like pyelonephritis was higher (Gagyor et al, 2015; Vik et al, 2018). Interestingly, neither nonselective COX inhibitors (i.e., ibuprofen) or selective COX-2 inhibitors (i.e., Celecoxib) have been evaluated in conjunction with antibiotics as adjunct therapies for UTI. Our finding that postmenopausal women with above median levels of urinary PGE$_2$ have a significantly elevated risk of rUTI relapse alongside mechanistic evidence provided by studies in murine models support the hypothesis that high levels of COX-2–mediated inflammation sensitize the bladder to recurrent infection. These findings could be used to inform the design of future clinical trials evaluating the efficacy of adjunct antibiotic and selective COX-2 inhibitor therapies for rUTI.

## Materials and Methods

All studies were performed between May 2018 and June 2020 following informed patient consent and institutional review board approval (STU 082010-016, STU 032016-006, MR 17-120).

### Cohort 1 and 2 biopsy and urine sample collection

Cold cup biopsies and urine were collected from postmenopausal women under anesthesia undergoing outpatient cystoscopy with fulguration of trigonitis (CFT) for advanced management of antibiotic-refractory rUTI. Exclusion criteria were immunodeficiency, renal insufficiency, urogenital abnormality, and any surgery 1 mo prior. Qualification for antibiotic-refractory rUTI requires presentation of >3 antibiotic class allergies and >3 antibiotic class resistances (Chavez et al, 2020). For Cohort 1, one C1 (no visible cystitis, control) and one I1 (visible cystitis, inflamed) biopsy was obtained (n = 14, PNK001-14) (De Nisco et al, 2019). Cohort 1 patient statistics have been previously reported (De Nisco et al, 2019). For Cohort 2, one I1 biopsy was collected (n = 12, PNK016-27). PNK021 did not pass exclusion criteria. Biopsies were fixed immediately in 4% paraformaldehyde, paraffin-embedded, and longitudinally sectioned using sterile solutions and equipment (De Nisco et al, 2019). Paraffin-embedded normal bladder tissue sections (HUFPT108) were purchased from US Biomax.

### Cohort 3 recruitment, urine collection, and classification

92 patients passing exclusion criteria of premenopausal, sporadic UTI, PVR > 150 ml, >stage 2 bladder prolapse, immune suppression, history of catheterization, and surgery less than a month prior were recruited from a tertiary care center into Cohort 3. Clean-catch midstream urine was collected, immediately chilled and processed within 2 h. Urine was handled aseptically and stored in liquid nitrogen. The 92 women were stratified into three groups based on their history of rUTI, UTI symptoms, and urinalysis (UA): Never (no clinical history of symptomatic UTI, n = 26), Remission (history of rUTI, no current UTI symptoms, and −UA, n = 35), and Relapse (history of rUTI, current UTI symptoms, and +UA, n = 31). The women in this cohort were distinct from cohorts 1 and 2 in that their rUTI was not yet refractory to antibiotics and they were therefore not

undergoing CFT. For nonsteroidal anti-inflammatory drug (NSAID) classification, patients taking ibuprofen, aspirin or naproxen were placed in the nonselective NSAID group and patients taking Meloxicam or Celecoxib were placed in the selective NSAID group.

### PGE$_2$ and creatinine measurement

Urinary PGE$_2$ levels were measured by highly sensitive PGE$_2$ ELISA (Enzo). To inhibit the activity of prostaglandin synthase, 10 µg/ml indomethacin (Alfa Aesar) was added. PGE$_2$ ELISA was performed on diluted urine (1:2) and standards. Optical density was measured at 405 nm with a Synergy H1 plate reader (BioTek). PGE$_2$ concentration was calculated based on the standard curve. Creatinine level was measured in diluted urine (1:20) and standards with the Creatinine Urinary Detection kit (Thermo Fisher Scientific). Absorbance was read at 490 nm and creatinine concentration was calculated based upon the standard curve.

### 16S rRNA FISH

16S rRNA FISH was performed as described previously (Neugent et al, 2019). Biopsies were fixed immediately upon collection and were processed using sterile reagents and aseptic technique. Briefly, after deparaffinization and rehydration, tissues were incubated with 10 nM Alexa Fluor-647–conjugated probe overnight at 50°C, washed and stained with 1 µg/ml Hoechst (Thermo Fisher Scientific), and mounted.

### Immunofluorescence (IF)

Tissues were deparaffinized and rehydrated as described previously (De Nisco et al, 2019). Following antigen retrieval in 10 mM citrate buffer, tissues were blocked in 5% goat serum (Krenacs et al, 2010). IF was performed with primary antibodies against COX-2 (D5H5) 1:500 (rabbit; Cell Signaling) and 2 µg/ml neutrophil elastase (ELA-2, 950334, mouse; Novus). Secondary antibodies Alexa Fluor-555 goat anti-rabbit IgG(H+L) and Alexa Fluor-647 goat anti-mouse IgG(H+L) (Thermo Fisher Scientific) were added to a final concentration of 4 and 2 µg/ml, respectively. Hoechst 33342 final concentration was 1 µg/ml. Alexa Fluor-488 phalloidin and Alexa Fluor-488 Wheat Germ Agglutinin were used at a 1:500 dilution.

### Imaging and analysis

For both FISH and IF, confocal micrographs were taken with 63× objectives on a Zeiss LSM880. Images were of a single focal plane and were processed and analyzed with Zen Blue (Zeiss) and ImageJ. 10 randomly sampled images were enumerated from each biopsy section in a blinded manner to calculate the percentage of COX-2 expressing cells and neutrophils present in the urothelium. Cells with a mean fluorescence intensity >10 and integral fluorescence density >1,000 in the Alexa Fluor-555 channel were scored as COX-2 positive. Cells with nuclei surrounded by ELA-2 signal were counted as neutrophils and neutrophils were distinguished from neutrophil extracellular traps by morphology (Brinkmann et al, 2004; Kaplan & Radic, 2012). For 16S rRNA FISH, bacterial communities (Universal 16s rRNA probe) were enumerated in a blinded manner in 10 randomly sampled images taken from a single section of each biopsy and compared with control (Scramble) images taken from the same region of a serial section as previously described (De Nisco et al, 2019).

### Time-to-relapse analysis

Time to relapse in the relapse group was assessed by Kaplan–Meier analysis. Patients were dichotomized about the median urinary PGE$_2$ concentration (n = 16 ≤ median, n = 15 > median). Median PGE$_2$ was chosen as the discriminator because it is an unbiased method for dichotomizing a group of individuals. Patients were censored at last follow-up time who did not complete a 12-mo follow-up interval (n = 3). The Mantel–Cox log-rank test was performed to test for a time-to-relapse difference between groups.

### Statistical analysis

All statistical analyses were performed in GraphPad Prism 8.1.0 and R Studio version 4.0.2 with an α of 0.05. Hypothesis testing was performed using classical methods. Pairwise associations between continuous variables were performed using Spearman's rho correlation with P-values generated by permutation. Differences between continuous variables by group were analyzed using ANOVA or t test if normally distributed, or the Mann–Whitney U test, Wilcoxon matched pairs signed rank test, or Kruskal–Wallis test with multiple comparison post hoc if not normally distributed. Differences between continuous variable by group were analyzed using ANOVA, t test, or Kruskal–Wallis test with multiple comparison post hoc. Differences between categorical variables were analyzed using chi-square, Fisher's exact test, or ordinal logistic regression. Various logistic regression models were constructed to predict rUTI using combinations of all significant covariates. Model fit was assessed using McFadden's pseudo-$R^2$. The predictive power of each logistic regression model was assessed using F-scores and AUC attained through leave-one-out cross validation. Models were further compared using ANOVA.

## Data Availability

All experimental source data are available upon request from the corresponding author.

## Supplementary Information

## Acknowledgements

This work was funded in part by the Felecia and John Cain Chair in Women's Health in honor of PE Zimmern, the Welch Foundation grant AT-2030-20200401 (NJ De Nisco), the University of Texas System Rising STARs award (NJ De Nisco), the Welch Foundation grant I-1561 (K Orth), and Once

Upon a Time Foundation (K Orth). K Orth is a Burroughs Welcome Investigator in Pathogenesis of Infectious Disease, a Beckman Young Investigator, and a W. W. Caruth, Jr., Biomedical Scholar and has an Earl A. Forsythe Chair in Biomedical Science.

## Author Contributions

T Ebrahimzadeh: formal analysis, investigation, visualization, methodology, and writing—original draft.

A Kuprasertkul: data curation, investigation, and methodology.

ML Neugent: formal analysis, visualization, methodology, and writing—original draft.

KC Lutz: formal analysis, visualization, methodology, and writing—original draft.

JL Fuentes: data curation and investigation.

J Gadhvi: investigation and visualization.

F Khan: investigation.

C Zhang: formal analysis.

BM Sharon: validation and investigation.

K Orth: funding acquisition and writing—review and editing.

Q Li: conceptualization, supervision, and validation.

PE Zimmern: conceptualization, supervision, funding acquisition, and writing—review and editing.

NJ De Nisco: conceptualization, supervision, funding acquisition, validation, methodology, writing—original draft, and project administration.

## Conflict of Interest Statement

The authors declare that they have no conflict of interest.

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
