## [Reviewer comments · Life Science Alliance]

Life Science Alliance

Urinary prostaglandin E2 as a biomarker for recurrent UTI in postmenopausal women

Tahmineh Ebrahimzadeh, Amy Kuprasertkul, Michael Neugent, Kevin Lutz, Jorge Fuentes, Jashkaran Gadhvi, Fatima Khan, Cong Zhang, Belle Sharon, Kim Orth, Qiwei Li, Philippe Zimmern, and Nicole De Nisco

DOI: <https://doi.org/10.26508/lsa.202000948>

Corresponding author(s): Nicole De Nisco, The University of Texas at Dallas

Review Timeline:	Submission Date:	2020-10-26
	Editorial Decision:	2020-11-30
	Revision Received:	2021-03-20
	Editorial Decision:	2021-04-13
	Revision Received:	2021-04-21
	Accepted:	2021-04-26

Scientific Editor: Shachi Bhatt

Transaction Report:

November 30, 2020

Re: Life Science Alliance manuscript #LSA-2020-00948-T

Dr. Nicole J De Nisco
The University of Texas at Dallas
Biological Sciences
800 W. Campbell Road
BSB12.515
Richardson, Texas 75080

Dear Dr. De Nisco,

Thank you for submitting your manuscript entitled "Urinary prostaglandin E2 is a biomarker for recurrent urinary tract infection in postmenopausal women" to Life Science Alliance. The manuscript was assessed by expert reviewers, whose comments are appended to this letter.

As you will note from the reviews below, all three reviewers find your work interesting, but have raised a few concerns and requested some clarifications that need to be addressed before we can consider the work any further. Thus, we encourage you to address all of the reviewers' points and re-submit the revised manuscript to us. While we agree with Rev2 that providing segregated data based on NSAID classification will be interesting for this study, we would understand if the sample sizes are limited to perform such an analysis. Performing such analysis is not a requirement to publish in LSA. Do let us know if you will be able to address this concern

Thank you for this interesting contribution to Life Science Alliance. We are looking forward to receiving your revised manuscript.

Sincerely,

Shachi Bhatt, Ph.D.
Executive Editor
Life Science Alliance
<https://www.lsjournal.org/>
Tweet @SciBhatt @LSAJournal

- A letter addressing the reviewers' comments point by point.
- An editable version of the final text (.DOC or .DOCX) is needed for copyediting (no PDFs).
- High-resolution figure, supplementary figure and video files uploaded as individual files: See our detailed guidelines for preparing your production-ready images, <https://www.life-science-alliance.org/authors>
- Summary blurb (enter in submission system): A short text summarizing in a single sentence the study (max. 200 characters including spaces). This text is used in conjunction with the titles of papers, hence should be informative and complementary to the title and running title. It should describe the context and significance of the findings for a general readership; it should be written in the present tense and refer to the work in the third person. Author names should not be mentioned.

B. MANUSCRIPT ORGANIZATION AND FORMATTING:

Reviewer #1 (Comments to the Authors (Required)):

Recurrent UTI is a pressing problem and constitutes a huge economic healthcare burden, as well as being an exacerbator of global antimicrobial resistance and the cause of undue morbidity. Like so many other afflictions more common in women than men, research into this area has been limited. Those studies that are done are almost always performed in mice, which are not always reliable

models. So it is good to see work in this important area on human cohorts. This study shows that urinary PGR2 can identify women with rUTI and can predict its severity. As much, it may prove quite useful for stratifying patients and deciding where prophylactic antibiotic use might be useful. It might also serve as the basis for a therapeutic approach. (Despite what I said above, about the occasional unreliability of mouse models, it should be noted that the key findings presented in this paper have already been shown in mice. It's good to confirm in humans but it does temper its originality somewhat.) I think this paper is a solid and interesting body of data, and the result of a lot of hard and careful work, which should be published. Just a few comments below.

1. Data in Fig 3 are not entirely supported.

Given that some uropathogens are thought to have intracellular phases, it's really nice to see biopsy FISH in a paper. I just have a few questions about this figure. Is what's shown just one slice of a confocal? Or a max projection? Or what? Are you able to show evidence of bacteria in multiple layers of the same area at different z-heights? In other words have you ruled out that these bacteria are not just clinging to the cut surface of the biopsy (potential contamination at some point in the pipeline)? Although the image is a bit small to say for sure, it seems a bit odd that all of the bacteria look perfectly in focus. Would be important to see your 3D analysis of the bacteria-associated region.

2. Does the data apply to all rUTI as implied by title/abstract?

Is the trigonitis cohort generalizable to other rUTI cohorts - the title, abstract and intro only talk about rUTI, and it's not until you reach the results that you see that it might be a particular "type" of rUTI. If you believe it does, this should probably be discussed in the introduction for those not familiar with patients who get referred for this particular procedure.

Additional minor issues

2. "Fig 1. Having no control biopsy (no disease) makes it harder to assess these stainings, although I see you've got a control in later figures. I was intrigued by the stark difference between the inflamed regions of 006 and 011 in Fig 1B. Why do you think this is? If 011 has 3-4% neutrophils (on the graph) is it surprising there isn't even one on the "representative" picture in B? It looks like at least 100 cells there. Perhaps choose a different representative image that has a few neutrophils in the micrograph...

3. Suburothelial" - define for those readers not experts in histology - are you talking about lamina propria, muscularis propria, adventitia, or where exactly? Nice to know without checking the supplemental data

4. Who is the control in Fig 2B? Is it one of the "Never" cohort? Need to specify. Minor point, you mention (c) in the legend to Fig 2B but that's not on the figure, it's spelled out.

5. Fig 2: what's up with patient 16? Sorry if I missed it but I was curious why she had very low COX/neutrophils but sky-high PGE2 levels and a positive MSU. Just generally, it would be nice to see a discussion about the various exceptions to the rule - why don't 100% have these biomarkers despite being take from/having very inflamed regions?

Reviewer #2 (Comments to the Authors (Required)):

The manuscript by Ebrahimzadeh et al extends previously published observations of cyclooxygenase-2 (COX-2)-mediated inflammation in the preclinical model of urinary tract infection (UTI) to the clinic. The strengths of the manuscript include the use of 3 different human cohorts, complementary microscopic and urinalysis approaches, the inclusion of both normalized and raw urinalysis results, the stratification of subjects and the longitudinal follow-up studies. The manuscript has some superficially presented sections, particularly in the methodology that make it difficult to interpret the authors findings and to replicate the studies. In addition, this subject population, while providing the best opportunity to obtain clinical biopsies, has significant limitations in the extrapolation of the findings to the general postmenopausal woman population. There are additional limitations to the study that should be indicated. The title is overstated and should be tempered.

Major concerns:

1. There is a difference in the antibiotic exposure prior to the cystoscopy with fulguration of trigonitis (CFT). There was no description as to why there was a difference. Is this a change in standard of care or was it to specifically address a hypothesis.
2. Subjects with CFT for advanced management of recurrent UTI is not standard of care and likely represents those subjects with the most recalcitrant recurrences. As such, there should be inclusion of the criterion for selection of CFT in the treatment. Also, the inference that the observations are relevant to all postmenopausal women is overstated and the language should be tempered.
3. Figure 1. The authors nicely obtain biopsies from women in areas that are visibly inflamed and those that are not from a cohort of subjects. The authors enumerate the number of cells that are positive for COX-2 production (not expression) and the number of neutrophils present. Additional experimental detail is needed. What was the minimal fluorescence level that was used to determine whether a cell was positive for COX-2? Since neutrophil elastase is secreted, how were the regions with large multi-lobular staining (as in the lower left region of PKN006-inflamed interpreted? How were the total number of urothelial cells enumerated, was this based upon counting nuclei? As with other studies, it is informative to the reader if the delineation of the urothelium from the sub-urothelium could be provided on the images. Please define "moderate to high" for the sub-urothelium observations. In these types of analyses, blinded random sampling of the tissue provides a more accurate analysis than the use of representative images, as the investigator is interpreting the data to obtain a representative, which can introduce bias. The inclusion of representative images for the figure is appropriate. Please consider a different color (or no color) for either the neutrophils or COX-2, the similarity between the red and magenta makes it difficult for some to distinguish. Were the 10 representative images from the same section of tissue or were multiple tissue sections analyzed? The use of multiple sections for this type of analysis provides additional rigor.
4. Figure 2. Please indicate what the control for the enumeration of COX-2 positive cells and neutrophils. The inclusion of raw and normalized urine levels is appreciated.
5. Figure 3. Please provide additional details on how the bacterial community size was determined. Please indicate how many sections were used to obtain the 10 images that were used in the analyses. In addition, documentation of sub-urothelial bacterial communities are limited, particularly in humans. Thus it would be of interest to provide this data separately as this would be of interest to the field.
6. Figure 4. Panel A in the legend indicates that workflow is depicted but is not present, please

revise. In the methods, the subjects in Cohort 3 were also segregated based upon the class of NSAID being actively taken by each subject. It would be interesting to also provide segregated data on the type of NSAID classification. This is of particular importance given the hypothesis for modulation of the COX-2 pathway in the management of UTI.

7. Figure 5. The focus on diabetes is important as this is a known modifier of UTI susceptibility. Which group(s) are included in control, are these the never only or do these also include those in remission? What is the rationale for the choice of median as the discriminator for relapse? Please indicate that the median was selected from Figure 4C. If the level of PGE2 alone is a sufficient predictor, then there should have been some level of UTI in the control group for those with values over 2000 pg/ml.

Minor concerns:

1. The observation that the highest levels of PGE2 were in the subjects with positive urine culture is interesting. It would be helpful to change the color of these bars to indicate this finding to the reader.

Reviewer #3 (Comments to the Authors (Required)):

In this manuscript, Ebrahimzadeh et al, describe the investigation of COX-2 and its product PGE2 in the bladder and urine of postmenopausal women who experience recurrent UTI. Recurrent UTI in this group is a serious issue and studies to better understand causes or to identify biomarkers are needed. The main finding is that PGE2 levels are predictive of recurrent UTI relapse in this patient population. This manuscript is straightforward, clearly written, and importantly, recognizes both the findings and limitations in the study populations to draw clear conclusions.

Minor concerns include

1. Additional emphasis that the studies described in lines 56-67 were performed exclusively in mice would help to drive home the take home message in this study.
2. It's unclear what is meant by an defined cohort (line 68) - does this mean well-described/annotated?
3. In Fig 1B, it should be marked "elastase or ELA2" rather than "neutrophil" as is done in Fig S.
4. Why were no control regions taken in cohort 2?
5. Commercially available normal bladder biopsy tissue is mentioned in the results (line 119) but not in the Materials and Methods.
6. Are the labels correct in Fig. 3A? There is a large signal in the "scramble" and very low signal in the PNK025 sample.
7. It would be interesting to know when COX-2 levels begin to increase with respect to the presence of bacteria and symptoms in patients. Do the authors have any data or information on this?
8. Why are remission patients not included in the AODM analyses?
9. Line 216 - if the results are not significant, please refrain from describing any differences.

We thank the reviewers for their thorough and thoughtful review of our manuscript. We have revised the manuscript considering the reviewer comments, and we thank the reviewers for helping us to improve the quality of the manuscript. We especially thank the reviewers for pointing out cases where our methods or cohort descriptions may have been unclear. We have taken special care to improve these aspects of the manuscript. Please see our point-by-point responses to each reviewer's comments below.

Reviewer #1 (Comments to the Authors (Required)):

Recurrent UTI is a pressing problem and constitutes a huge economic healthcare burden, as well as being an exacerbator of global antimicrobial resistance and the cause of undue morbidity. Like so many other afflictions more common in women than men, research into this area has been limited. Those studies that are done are almost always performed in mice, which are not always reliable models. So it is good to see work in this important area on human cohorts. This study shows that urinary PGR2 can identify women with rUTI and can predict its severity. As much, it may prove quite useful for stratifying patients and deciding where prophylactic antibiotic use might be useful. It might also serve as the basis for a therapeutic approach. (Despite what I said above, about the occasional unreliability of mouse models, it should be noted that the key findings presented in this paper have already been shown in mice. It's good to confirm in humans but it does temper its originality somewhat.) I think this paper is a solid and interesting body of data, and the result of a lot of hard and careful work, which should be published. Just a few comments below.

1. Data in Fig 3 are not entirely supported.

Given that some uropathogens are thought to have intracellular phases, it's really nice to see biopsy FISH in a paper. I just have a few questions about this figure. Is what's shown just one slice of a confocal? Or a max projection? Or what? Are you able to show evidence of bacteria in multiple layers of the same area at different z-heights? In other words have you ruled out that these bacteria are not just clinging to the cut surface of the biopsy (potential contamination at some point in the pipeline)? Although the image is a bit small to say for sure, it seems a bit odd that all of the bacteria look perfectly in focus. Would be important to see your 3D analysis of the bacteria-associated region.

Author response: We appreciate your insightful comments and feedback. The images are one slice of confocal and not max projections. We have established in our previous publications on this method which include a JoVE protocol that these tissue-associated bacteria are not contaminants ([doi: 10.1016/j.jmb.2019.04.008](https://doi.org/10.1016/j.jmb.2019.04.008), DOI: [doi:10.3791/60458](https://doi.org/10.3791/60458)). We maintain a sterile environment throughout our entire procedure of tissue preparation and processing. Biopsies were directly deposited into the sterile paraformaldehyde fixative (this happened in the operating room), washed in sterile PBS, and processed on sterile equipment using aseptic technique. Because the biopsies were fixed whole and immediately, only bacteria that were present within the tissue would be fixed in the tissue. Importantly, as noted in our previous publications, we do not see tissue-associated bacteria in every biopsy or even in biopsies taken

from different bladder regions of the same patient and processed on the same day. We routinely see presence of bacteria in one biopsy and not the other taken from the same patient. If the presence of bacteria was due to contamination during the preparation, we should have observed presence of bacteria in all biopsy sections processed by that method on that day. Furthermore, our observations are consistent between serial sections. If we see tissue-associated bacteria in one biopsy section, we see it in all of the subsequent sections we take from that biopsy and vice versa. The bacteria are in focus because the image was taken using confocal microscopy which enabled us to focus on bacteria that were in the same Z-plane. We believe that the request for 3D analysis is beyond the scope of this manuscript because the focus of this paper is prostaglandin E2 and its utility as a rUTI biomarker and not the subcellular localization tissue-associated bacteria. We used our FISH method, which has already been described in the literature (by us and previously in the work it was adapted from), to demonstrate that in the absence of bacterial CFUs in the urine, the load of bacteria within the tissue may still be high. This observation is not essential to the major conclusions of this paper which are that urinary PGE₂ concentration is strongly associated with active rUTI and women with higher urinary PGE₂ have a greater risk of rUTI relapse. The purpose of this paper was not to characterize the tissue-associated bacterial communities. To determine where the bacteria are residing in these tissues would require immunofluorescence microscopy staining for specific cellular markers to determine co-localization. This is the subject of a separate manuscript and is out of the scope of this manuscript.

2. Does the data apply to all rUTI as implied by title/abstract?

Is the trigonitis cohort generalizable to other rUTI cohorts - the title, abstract and intro only talk about rUTI, and it's not until you reach the results that you see that it might be a particular "type" of rUTI. If you believe it does, this should probably be discussed in the introduction for those not familiar with patients who get referred for this particular procedure.

Author response: We appreciate this point, and we apologize if our cohort descriptions were unclear. We do believe that our findings are applicable to rUTI in general in postmenopausal women. While the two cohorts we used in the biopsy studies were certainly advanced in their disease progression, the urinary biomarker analysis including the time to relapse analysis was performed using cohort 3. This cohort consisted of three groups of women: those who had never experienced a symptomatic UTI, those who had recent rUTI history but were currently not experiencing infection, and those with rUTI history who were currently experiencing UTI. None of these patients were undergoing CFT at the time and the women in the active rUTI group are representative of Dr. Zimmern's patient population of postmenopausal women which includes women at all stages of the diseases with and without trigonitis. We believe that the third cohort makes the conclusions about PGE₂ work generalizable for postmenopausal rUTI patients. We have included a more detailed explanation of cohort 3 in the material and methods section to clarify this point (lines 315-317). We have also clarified this by adding the following text to the results and discussion section (lines 187-189):

"None of the women in Cohort 3 were undergoing CFT and active rUTI in the Relapse group was managed by antibiotic therapy."

Additional minor issues

2. "Fig 1. Having no control biopsy (no disease) makes it harder to assess these stainings, although I see you've got a control in later figures. I was intrigued by the stark difference between the inflamed regions of 006 and 011 in Fig 1B. Why do you think this is? If 011 has 3-4% neutrophils (on the graph) is it surprising there isn't even one on the "representative" picture in B? It looks like at least 100 cells there. Perhaps choose a different representative image that has a few neutrophils in the micrograph..."

Author response: The cohort assessed in figure 1 had one biopsy taken from a visibly inflamed region and one biopsy taken from a visibly normal region to serve as an internal control. However, in some patients the entire bladder was inflamed (pancystitis) and no control region was available. The control in Fig2B is a commercially available normal human biopsy purchased from US Biomax that was used because we did not have internal control biopsies from the PNK016-PNK027 cohort. We think the stark difference between 006 and 011 is due to bacteriuria. As reported in [doi: 10.1016/j.jmb.2019.04.008](https://doi.org/10.1016/j.jmb.2019.04.008), at the time of biopsy PNK006 had 2×10^8 CFUs of *E. coli* in her urine and PNK011 had a negative urine culture. As the results in figure 2 suggest, we hypothesize that a higher load of uropathogenic bacteria in the urine or bladder wall is associated with higher levels of urinary PGE₂ and urothelial COX-2 expression. This hypothesis is what led us to investigate urinary PGE₂ as a diagnostic biomarker of active rUTI using cohort 3. As for the representative image used in 1B, we chose representative images based on COX-2 expression, but we have included an additional representative image for PNK011 that contain neutrophils in Figure S1 at the reviewer's request.

3. Suburothelial" - define for those readers not experts in histology - are you talking about lamina propria, muscularis propria, adventitia, or where exactly? Nice to know without checking the supplemental data

Author response: Thank you for pointing this out. We have included a definition of suburothelial in the text (lines 107-108) as per your suggestion. In most cases the suburothelial area referred to is the lamina propria but in some cases the muscularis propria was also included in the biopsy.

4. Who is the control in Fig 2B? Is it one of the "Never" cohort? Need to specify. Minor point, you mention (c) in the legend to Fig 2B but that's not on the figure, it's spelled out.

Author response: Thank you for this comment. The control in Fig 2B is a commercially available normal human biopsy purchased from US Biomax. Please refer to lines 304-305 in the materials and methods section. We have fixed the Fig 2B legend to match the figure.

5. Fig 2: what's up with patient 16? Sorry if I missed it but I was curious why she had very low COX/neutrophils but sky-high PGE₂ levels and a positive MSU. Just generally, it would be nice to see a discussion about the various exceptions to the rule - why don't 100% have these biomarkers despite being take from/having very inflamed regions?

Author response: This is a very thoughtful question and important point. Biopsies are a sampling of different visibly inflamed regions of the bladder taken at the clinician's discretion based on visible cues. There is the possibility that the region the biopsy was taken from was not representative of the inflamed regions or status of the entire bladder. This is why it is important to define inflammatory biomarkers that are not affected by sampling bias or by chance. This is why we sought to evaluate urinary PGE₂, which is the product of the COX-2 enzyme, as a biomarker for active rUTI using cohort 3 which contained cases (relapse) and controls (never). It is possible, in the case of PNK016, that the biopsy was not representative of the bladder and the regions expressing COX-2 were missed. Since taking a large number of biopsies increases risk to the patient, we sought to evaluate urinary PGE₂ as a less invasive and possibly more reliable method of evaluating bladder inflammatory status. The possible explanation for PNK016 was added to the result and discussion section lines 149-156.

“Although the urinary PGE₂ concentration of PNK016 was high (2670.7 pg/mL), the percentage of COX-2-positive urothelial cells and neutrophils was relatively low in the biopsied tissue (Fig. 2C,D). Since the selection of inflamed region biopsied was based on visible cues, one possible explanation is that the inflamed region biopsied for this patient was not representative of the bladder as a whole and regions with high numbers of COX-2-positive cells or neutrophils were not captured in the biopsy. For this reason, analysis of PGE₂ concentrations in the urine, which is less prone to sampling bias, is important.”

Reviewer #2 (Comments to the Authors (Required)):

The manuscript by Ebrahimzadeh et al extends previously published observations of cyclooxygenase-2 (COX-2)-mediated inflammation in the preclinical model of urinary tract infection (UTI) to the clinic. The strengths of the manuscript include the use of 3 different human cohorts, complementary microscopic and urinalysis approaches, the inclusion of both normalized and raw urinalysis results, the stratification of subjects and the longitudinal follow-up studies. The manuscript has some superficially presented sections, particularly in the methodology that make it difficult to interpret the authors findings and to replicate the studies. In addition, this subject population, while providing the best opportunity to obtain clinical biopsies, has significant limitations in the extrapolation of the findings to the general postmenopausal woman population. There are additional limitations to the study that should be indicated. The title is overstated and should be tempered.

Major concerns:

1. There is a difference in the antibiotic exposure prior to the cystoscopy with fulguration of trigonitis (CFT). There was no description as to why there was a difference. Is this a change in standard of care or was it to specifically address a hypothesis.

Author response: The change in antibiotic exposure between cohorts 1 and 2 was due to a change in the protocol requested by the IRB upon renewal of the protocol approval.

2. Subjects with CFT for advanced management of recurrent UTI is not standard of care and likely represents those subjects with the most recalcitrant recurrences. As such, there should be inclusion of the criterion for selection of CFT in the treatment. Also, the inference that the observations are relevant to all postmenopausal women is overstated and the language should be tempered.

Author response: CFT was elected by the patients in cohorts 1 and 2 as an option to treat antibiotic-refractory rUTI. We have included the selection criteria for this procedure in the methods section (please see line 296-299). However, cohort 3 did not include patients undergoing CFT and consisted of three groups: postmenopausal women with no history symptomatic UTI, postmenopausal women with recent history of rUTI without current symptomatic rUTI, and postmenopausal women with rUTI history with current symptomatic rUTI. The urinary PGE₂ biomarker study was performed using this cohort which is representative of postmenopausal women with rUTI in general. The majority of Dr. Zimmern's rUTI patient base is managed with antibiotic therapy and does not elect CFT. Therefore, we believe that because the biomarker analysis was performed using cohort 3 who did not elect CFT, our conclusions that PGE₂ is a diagnostic and prognostic biomarker for rUTI are relevant to postmenopausal women in general. We apologize if this description of cohort 3 was unclear and have added text to clarify this description in the methods section (lines 315-317):

"The women in this cohort were distinct from cohorts 1 and 2 in that their rUTI was not yet refractory to antibiotics and they were therefore not undergoing CFT."

3. Figure 1. The authors nicely obtain biopsies from women in areas that are visibly inflamed and those that are not from a cohort of subjects. The authors enumerate the number of cells that are positive for COX-2 production (not expression) and the number of neutrophils present. Additional experimental detail is needed. What was the minimal fluorescence level that was used to determine whether a cell was positive for COX-2? Since neutrophil elastase is secreted, how were the regions with large multi-lobular staining (as in the lower left region of PKN006-inflamed interpreted? How were the total number of urothelial cells enumerated, was this based upon counting nuclei? As with other studies, it is informative to the reader if the delineation of the urothelium from the sub-urothelium could be provided on the images. Please define "moderate to high" for the sub-urothelium observations. In these types of analyses, blinded random sampling of the tissue provides a more accurate analysis than the use of representative images, as the investigator is interpreting the data to obtain a representative, which can introduce bias. The inclusion of representative images for the figure is appropriate. Please consider a different color (or no color) for either the neutrophils or COX-2, the similarity between the red and magenta makes it difficult for some to distinguish. Were the 10 representative images from the same section of tissue or were multiple tissue sections analyzed? The use of multiple sections for this type of analysis provides additional rigor.

Author response: Thank you for this comment, we apologize if the description of our quantitation methods was lacking in detail. We have added text to the methods section to

provide the detail requested. Briefly, 10 randomly sampled micrographs from each biopsy section were scored. The scoring was performed blinded by a separate individual than the one who generated the micrographs. Representative images were only used for the figures. We apologize if **our** wording suggested that representative images were scored. The scoring of COX-2 positive cells was performed using a cutoff of mean fluorescence intensity >10 and integral density >1000. The total number of urothelial cells was enumerated by counting nuclei as stained with Hoechst. For neutrophils, the ELA2 antibody does not detect diffuse neutrophil elastase and cellular elastase was differentiated from neutrophil extracellular traps (NETs) by morphology ([DOI: 10.1126/science.1092385](https://doi.org/10.1126/science.1092385)). We counted a cell as a neutrophil if we observed a Hoechst-stained nucleus surrounded by ELA2 staining whereas DNA and elastase in NETs appears as distinctive strands ([doi: 10.4049/jimmunol.1201719](https://doi.org/10.4049/jimmunol.1201719)). The 10 representative images were from the same section because due to the small size of the biopsies the sections available for experimentation were limited. However, all sections analyzed were from the same level. We have added the following language to clarify our scoring methods in the materials and methods section (lines 346-352):

*“10 randomly sampled images were enumerated from each biopsy section in a blinded manner to calculate the percentage of COX-2 expressing cells and neutrophils present in the urothelium. Cells with a mean fluorescence intensity >10 and integral fluorescence density >1000 in the AlexaFluor-555 channel were scored as COX-2 positive. Cells with **nucelli** surrounded by ELA-2 signal were counted as neutrophils and neutrophils were distinguished from neutrophil extracellular traps by morphology”*

Thank you for the suggestion to delineate the suburothelium, we have included this in the figure images. As for changing the colors of the COX-2 and/or ELA-2 channel colors, we prefer to not false color the images but for ease of interpretation we have provided a figure panel of the micrographs with split COX-2 and ELA-2 channels in Figures S1 and S2. We also removed the terms “moderate to high” describing sub-urothelial neutrophil accumulation since it was not quantified (please see line 110).

4. Figure 2. Please indicate what the control for the enumeration of COX-2 positive cells and neutrophils. The inclusion of raw and normalized urine levels is appreciated.

Author response: Due to a change in the IRB-approved protocol, we only were able to biopsy the inflamed area for cohort 2. The control data in Figure 2 were collected from commercially available normal adult bladder biopsy sections purchased from US Biomax. We have added text to the figure 2 legend and to the methods section (please see lines 304-305) to clarify this.

5. Figure 3. Please provide additional details on how the bacterial community size was determined. Please indicate how many sections were used to obtain the 10 images that were used in the analyses. In addition, documentation of sub-urothelial bacterial communities are limited, particularly in humans. Thus it would be of interest to provide this data separately as this would be of interest to the field.

Author response: Thank you for your suggestions. Bacterial community size was enumerated as previously described in [doi: 10.1016/j.jmb.2019.04.008](https://doi.org/10.1016/j.jmb.2019.04.008), DOI: [doi:10.3791/60458](https://doi.org/10.3791/60458). We stained serial sections (e.g. L2 and L3) with either the 16S rRNA or the scramble probe and obtained micrographs from matching randomly sampled areas of each section. Bacterial communities were enumerated in each 16S rRNA image using the matching scramble images to discount background signal in a blinded fashion. Due to the small size of the biopsy, only two sections were available for FISH experiments. These sections are longitudinal with each one including the urothelium, lamina propria and sometimes muscularis propria. We have added the following text to the methods section to clarify the enumeration methods (lines 352-356).

“For 16S rRNA FISH, bacterial communities (Universal 16s rRNA probe) were enumerated in a blinded manner in 10 randomly sampled images taken from a single section of each biopsy and compared to control (Scramble) images taken from the same region of a serial section as previously described (De Nisco et al., 2019).”

We thank you for your interest in the suburothelial bacterial communities that we have documented in this manuscript and in our previous manuscript ([doi: 10.1016/j.jmb.2019.04.008](https://doi.org/10.1016/j.jmb.2019.04.008)). Although the bladder-associated bacterial communities were not the focus of this article, we have provided an additional supplemental figure containing micrographs of the suburothelial communities at the reviewer’s request (Fig S4). Additional analysis of these communities is outside of the scope of this manuscript as the manuscript’s primary focus is evaluating COX-2-mediated inflammation and urinary PGE₂ as a biomarker for rUTI in postmenopausal women.

6. Figure 4. Panel A in the legend indicates that workflow is depicted but is not present, please revise. In the methods, the subjects in Cohort 3 were also segregated based upon the class of NSAID being actively taken by each subject. It would be interesting to also provide segregated data on the type of NSAID classification. This is of particular importance given the hypothesis for modulation of the COX-2 pathway in the management of UTI.

Author response: Thank you for catching this error. We have revised the figure 4 legend to remove the word “workflow”. As per the author’s request we provided information of the type of NSAIDs taken by women in cohort 3 as a new panel in Figure S7. Please see Figure S7B for this information. The majority of patients taking non-selective NSAIDs were taking aspirin and the majority of patients taking selective NSAIDs were taking Meloxicam.

7. Figure 5. The focus on diabetes is important as this is a known modifier of UTI susceptibility. Which group(s) are included in control, are these the never only or do these also include those in remission? What is the rationale for the choice of median as the discriminator for relapse? Please indicate that the median was selected from Figure 4C. If the level of PGE₂ alone is a sufficient predictor, then there should have been some level of UTI in the control group for those with values over 2000 pg/ml.

Thank you for this comment. In Figure 5A, the control group consisted of both Never UTI and rUTI remission patients since neither were experiencing current UTI and we determined that there was no significant difference in the median or mean urinary PGE₂ concentration between the Never and Remission groups. We therefore chose to compare the trends of urinary PGE₂ concentration between patients not experiencing UTI (control) and patients experiencing UTI (relapse) in Non-diabetic versus diabetic patients. Median was chosen as the discriminator for the time to relapse analysis because it is an unbiased method for dichotomizing a group of individuals. To clarify, median was not chosen as a discriminator of relapse but just as a method to dichotomize the Relapse group so that we could compare outcomes of patients with above median and below median urinary PGE₂ concentrations. From this analysis, we are not attempting to report at a specific urinary PGE₂ concentration that is indicative of relapse just a difference in risk between patients with higher (above median) versus lower (below median) urinary concentrations of PGE₂. Determination of an exact urinary PGE₂ threshold that is indicative a relapse would require a much larger cohort of individuals and is beyond the scope of this manuscript.

Minor concerns:

1. The observation that the highest levels of PGE₂ were in the subjects with positive urine culture is interesting. It would be helpful to change the color of these bars to indicate this finding to the reader.

Author response: Thank you for this comment, we have changed the color of the bars for patients with positive urine to red in Figure 2F.

Reviewer #3 (Comments to the Authors (Required)):

In this manuscript, Ebrahimzadeh et al, describe the investigation of COX-2 and its product PGE₂ in the bladder and urine of postmenopausal women who experience recurrent UTI. Recurrent UTI in this group is a serious issue and studies to better understand causes or to identify biomarkers are needed. The main finding is that PGE₂ levels are predictive of recurrent UTI relapse in this patient population. This manuscript is straightforward, clearly written, and importantly, recognizes both the findings and limitations in the study populations to draw clear conclusions.

Minor concerns include

1. Additional emphasis that the studies described in lines 56-67 were performed exclusively in mice would help to drive home the take home message in this study.

Author response: Thank you for this feedback. We have noted this in Line 59.

2. It's unclear what is meant by an defined cohort (line 68) - does this mean well-described/annotated?

Author response: We mean that it has been heavily curated. Patients passed strict inclusion criteria and their clinical history has been thoroughly documented.

3. In Fig 1B, it should be marked "elastase or ELA2" rather than "neutrophil" as is done in Fig S.

Author response: Thank you for pointing this out. We have updated figures 1B and 2B accordingly.

4. Why were no control regions taken in cohort 2?

Author response: Upon resubmission of our protocol for IRB approval, we made changes to the protocol in an effort to further minimize the risk to our patients. These included no longer taking biopsies from regions that were not to be fulgurated (control regions) and not taking the patients off of antibiotic therapy prior to CFT. This is why we separated cohort 1 and cohort 2 in our analysis. Also, in lieu of a matched control region we used commercially available normal human bladder biopsy sections as a comparator for cohort 2 (please see the materials and methods section line 305-306).

5. Commercially available normal bladder biopsy tissue is mentioned in the results (line 119) but not in the Materials and Methods.

Author response: Thank you for pointing this out. Please see the methods section lines 304-05.

6. Are the labels correct in Fig. 3A? There is a large signal in the "scramble" and very low signal in the PNK025 sample.

Author response: Yes, the two left panels are the 16S rRNA universal probe and the right two panels are the scramble probe. Positive FISH samples are characterized by small, bright signals denoted by the white arrowheads. The signal in PNK027 scramble panel (bottom right) is non-specific/background signal. This is why it is critical to hybridize a serial section with the scramble probe because it allows determination of specific versus non-specific signal.

7. It would be interesting to know when COX-2 levels begin to increase with respect to the presence of bacteria and symptoms in patients. Do the authors have any data or information on this?

Author response: Yes, we agree that this would be very interesting! We are currently performing a longitudinal study to answer this question. However, the cross-sectional study in this manuscript cannot address that question.

8. Why are remission patients not included in the AODM analyses?

Author response: For the AODM analysis we stratified patients based on current UTI status and AODM diagnosis. The control group consisted of women without UTI at time of sample collection (both Never UTI and remission) and the relapse group consisted of women with active UTI at the time of sample collection. We chose to do this because our data suggest that elevated urinary PGE₂ concentration is associated with active UTI and there was no detectable difference in urinary PGE₂ concentrations between women with and without rUTI history who were not currently experiencing an active UTI (Fig. 4B,C). We realize now that the designation of control lacks clarity and have thus revised the text and figure to read “No active UTI” in lieu of control and “Active UTI” in lieu of Relapse.

9. Line 216 - if the results are not significant, please refrain from describing any differences.

Author response: Thank you for your input we have removed the description of these data.

April 13, 2021

RE: Life Science Alliance Manuscript #LSA-2020-00948-TR

Dr. Nicole J De Nisco
The University of Texas at Dallas
Biological Sciences
800 W. Campbell Road
BSB12.515
Richardson, Texas 75080

Dear Dr. De Nisco,

Thank you for submitting your revised manuscript entitled "Urinary prostaglandin E2 as a biomarker for recurrent UTI in postmenopausal women". We would be happy to publish your paper in Life Science Alliance pending minor text revision as requested by Reviewer 1 and final revisions necessary to meet our formatting guidelines.

Along with the points mentioned below, please also attend to the following:

- please consult our manuscript preparation guidelines <https://www.life-science-alliance.org/manuscript-prep> and make sure your manuscript sections are in the correct order;
- please separate the Results and Discussion section into two - 1. Results 2. Discussion, as per our formatting requirements
- please add your main, supplementary figures, and table legends to the main manuscript text, after the references section
- please separate the Figure legends and Supplemental Table legends into separate sections
- we encourage you to revise the figure legends for figures S1, S4, S6 such that the figure panels are introduced in an alphabetical order
- please add a callout for Figure S7 to your main manuscript text
- The panels in Figure 1B and S1 are the same. Also, panels in Figure 2B and S3 are the same. It is our policy to not allow for figure duplications in the manuscript. Would it be possible to use a different image in either of those figures? If it is necessary to use the same image, we would ask to clarify that the panels have been repeated in the figure legend.

A. FINAL FILES:

B. MANUSCRIPT ORGANIZATION AND FORMATTING:

Sincerely,

Shachi Bhatt, Ph.D.
Executive Editor
Life Science Alliance
<http://www.lsajournal.org>
Tweet @SciBhatt @LSAJournal

Reviewer #1 (Comments to the Authors (Required)):

I have read through all of the author responses to all three referees and I think they have done a good job addressing the comments - we can agree to disagree about what is beyond the scope. My only residual concern is where the authors have rebutted a comment but have made no changes to the manuscript to make things clearer. If a referee had a question, it's likely readers might too - but readers will not have access to the authors' explanations. I personally think before this can be published, *all* responses should be accompanied by a change to the manuscript (even if it's very minor).

Reviewer #2 (Comments to the Authors (Required)):

The authors have respectfully revised the manuscript to my satisfaction. I have no additional concerns.

Reviewer #3 (Comments to the Authors (Required)):

Having read the reviewer comments, the authors' responses, and the revised manuscript, I am satisfied that any concerns raised have been met. I maintain that this manuscript is straightforward, clearly written, and importantly, recognizes both the findings and limitations in the study populations to draw clear conclusions. I think it brings important information to this field and look forward to seeing it published.

Response to editorial and reviewer comments

Life Science Alliance – LSA-2020-00948-TRR (minor revision)

Thank you for your careful review of our manuscript. We have edited the manuscript in response to the request for minor revision by the editor and reviewer 1. Please find our response to editorial and reviewer comments below:

Editorial requests:

-please consult our manuscript preparation guidelines and make sure your manuscript sections are in the correct order

We have reorganized the manuscript according to the Life Science Alliance manuscript preparation guidelines

-please separate the Results and Discussion section into two - 1. Results 2. Discussion, as per our formatting requirements

We have separated the Results and Discussion

-please add your main, supplementary figures, and table legends to the main manuscript text, after the references section

We have added the main text figure legends, supplementary figure legends, tables and their legends to the manuscript after the references section in accordance with the Life Science Alliance manuscript preparation guidelines

-please separate the Figure legends and Supplemental Table legends into separate sections

We have separated these sections according to Life Science Alliance manuscript preparation guidelines

-we encourage you to revise the figure legends for figures S1, S4, S6 such that the figure panels are introduced in an alphabetical order

We have revised the figure legends for S1 and S4 such that figure panels are introduced in alphabetical order. Figure S6 is just a single panel.

-please add a callout for Figure S7 to your main manuscript text

We have added callouts to Figure S7 to the main manuscript text. Thank you for catching this oversight.

-The panels in Figure 1B and S1 are the same. Also, panels in Figure 2B and S3 are the same. It is our policy to not allow for figure duplications in the manuscript. Would it be possible to use a different image in either of those figures? If it is necessary to use

the same image, we would ask to clarify that the panels have been repeated in the figure legend.

We included these images at the reviewer request include split channels in order to make the images more accessible to individuals who may be colorblind. We have clarified that these are repeated images from Figure 1B and Figure 2B in the legend and that they are provided for ease of interpretation.

Reviewer 1 request:

My only residual concern is where the authors have rebutted a comment but have made no changes to the manuscript to make things clearer. If a referee had a question, it's likely readers might too - but readers will not have access to the authors' explanations. I personally think before this can be published, *all* responses should be accompanied by a change to the manuscript (even if it's very minor).

As requested by reviewer 1 we have added textual changes to the manuscript in response to the few reviewer comments that were not accompanied by textual changes in the first round of review. Please see the list of changes below:

Reviewer 1, comment 1:

Is what's shown just one slice of a confocal? Or a max projection? Or what?

To clarify this point we added the following to line 363 in the methods section:

"Images were of a single focal plane and were processed and analyzed with Zen Blue (Zeiss) and ImageJ"

We have also added the following text to the FISH methods (line 350-351) in response to the reviewer's concern about sterility:

"Biopsies were fixed immediately upon collection and were processed using sterile reagents and aseptic technique."

Reviewer 1, minor comment 2:

I was intrigued by the stark difference between the inflamed regions of 006 and 011 in Fig 1B.

We added the following text to the results section (lines 98-100) in response to this reviewer comment:

"COX-2-positive urothelial cells were not observed in every I1 biopsy. Representative images of I1 biopsies with high urothelial COX-2 (PNK006) versus undetectable urothelial COX-2 (PNK011) are presented in Figure 1B."

Reviewer 2, minor comment 1:

There is a difference in the antibiotic exposure prior to the cystoscopy with fulguration of trigonitis (CFT). There was no description as to why there was a difference. Is this a change in standard of care or was it to specifically address a hypothesis.

We added the following text on line 127 for clarification in response to this comment:

"For this cohort, due to changes in the IRB-approved protocol, antibiotic therapy was not ceased prior to CFT...."

Reviewer 2, comment 7:

We previously added textual changes to address all questions posed (changes indicated in response to other comments that were redundant) in this comment except:

What is the rationale for the choice of median as the discriminator for relapse?

We added the following text in line 379-380 of the methods to clarify this point:

“Median PGE₂ was chosen as the discriminator because it is an unbiased method for dichotomizing a group of individuals.”

Reviewer 3, minor comment 1:

It's unclear what is meant by an defined cohort (line 68) - does this mean well-described/annotated?

We added the following text to line 72 for clarification:

“Here, we use defined, well-curated human cohorts to evaluate activation....”

April 26, 2021

RE: Life Science Alliance Manuscript #LSA-2020-00948-TRR

Dr. Nicole J De Nisco
The University of Texas at Dallas
Biological Sciences
800 W. Campbell Road
BSB12.515
Richardson, Texas 75080

Dear Dr. De Nisco,

Thank you for submitting your Research Article entitled "Urinary prostaglandin E2 as a biomarker for recurrent UTI in postmenopausal women". It is a pleasure to let you know that your manuscript is now accepted for publication in Life Science Alliance. Congratulations on this interesting work.

DISTRIBUTION OF MATERIALS:

Again, congratulations on a very nice paper. I hope you found the review process to be constructive and are pleased with how the manuscript was handled editorially. We look forward to future exciting submissions from your lab.

Sincerely,

Shachi Bhatt, Ph.D.

Executive Editor

Life Science Alliance

<http://www.lsjournal.org>
